# The Fungicidal Action of Micafungin is Independent on Both Oxidative Stress Generation and HOG Pathway Signaling in *Candida albicans*

**DOI:** 10.3390/microorganisms8121867

**Published:** 2020-11-26

**Authors:** Rebeca Alonso-Monge, José P. Guirao-Abad, Ruth Sánchez-Fresneda, Jesús Pla, Genoveva Yagüe, Juan Carlos Argüelles

**Affiliations:** 1Departamento de Microbiología y Parasitología-IRYCIS, Unidad de Microbiología, Facultad de Farmacia, Universidad Complutense de Madrid, E-28040 Madrid, Spain; guiraojo@ucmail.uc.edu (J.P.G.-A.); jpla@ucm.es (J.P.); 2Área de Microbiología, Facultad de Biología, Universidad de Murcia, E-30071 Murcia, Spain; ruth.sanchez1@um.es; 3Servicio de Microbiología Clínica, Hospital Universitario Virgen de la Arrixaca, IMIB, 30120 Murcia, Spain; gyague@um.es

**Keywords:** HOG pathway, antifungals, amphotericin B (AMB), micafungin (MF), MAPK phosphorylation, ROS, *Candida albicans*

## Abstract

In fungi, the Mitogen-Activated Protein kinase (MAPK) pathways sense a wide variety of environmental stimuli, leading to cell adaptation and survival. The HOG pathway plays an essential role in the pathobiology of *Candida albicans*, including the colonization of the gastrointestinal tract in a mouse model, virulence, and response to stress. Here, we examined the role of Hog1 in the *C. albicans* response to the clinically relevant antifungal Micafungin (MF), whose minimum inhibitory concentration (MIC) was identical in the parental strain (RM100) and in the isogenic homozygous mutant *hog1* (0.016 mg/L). The cell viability was impaired without significant differences between the parental strain, the isogenic *hog1* mutant, and the Hog1^+^ reintegrant. This phenotype was quite similar in a collection of *hog1* mutants constructed in a different *C. albicans* background. MF-treated cells failed to induce a relevant increase of both reactive oxygen species (ROS) formation and activation of the mitochondrial membrane potential in parental and *hog1* cells. MF was also unable to trigger any significant activation of the genes coding for the antioxidant activities catalase (*CAT1*) and superoxide dismutase (*SOD2*), as well as on the corresponding enzymatic activities, whereas a clear induction was observed in the presence of Amphotericin B (AMB), introduced as a positive control of Hog1 signaling. Furthermore, Hog1 was not phosphorylated by the addition of MF, but, notably, this echinocandin caused Mkc1 phosphorylation. Our results strongly suggest that the toxic effect of MF on *C. albicans* cells is not mediated by the Hog1 MAPK and is independent of the generation of an internal oxidative stress in *C. albicans*.

## 1. Introduction

*Candida albicans* is an inhabitant of the human microbiota with a huge potential to cause damage. This fungus causes superficial infections that can affect almost any mucosal surface of the body. Although the incidence of these infections is certainly elevated, more worrying are the systemic infections, especially in immunocompromised individuals [1,2,3]. *C. albicans* is still the most prevalent fungal human pathogen, being considered an important etiological cause of life-threatening fungal infections [4,5,6,7,8]. The armamentarium of antifungal compounds currently available for clinical therapy is limited, being polyenes (Amphotericin B) and azoles the most widely used compounds. The cellular target of Amphotericin B (AMB) is the ergosterol present in the fungal plasma and mitochondrial membranes. AMB binds ergosterol, forming pores and altering selective permeability [9,10]. On the other hand, azoles halt ergosterol biosynthesis by the specific inhibiting of 14α-demethylase (Erg11), leading to the depletion of ergosterol and accumulation of toxic sterol precursors [10]. The clinical introduction of echinocandins opened new promising therapeutic options, since this antifungal family acts through a novel fungicidal mechanism, i.e., the inhibition of β-(1,3)-glucan synthesis, a structural component of the fungal cell wall that is absent in mammal cells. However, whereas the resistance to AMB remains episodic more than 60 years after its introduction in the medical practice [11], the increasing isolation of strains resistant to azoles and echinocandins raises new difficulties in antifungal chemotherapy, considering particularly hospital-growing opportunistic infections [12,13]. Resistance to echinocandins is mainly due to mutations on “hot spot” regions of *FKS1* and *FKS2* genes that encode the β-(1,3)-D-glucan synthase [14], while a resistance to azoles involves other different and non exclusive mechanisms [15]. In *Candida albicans* and other *Candida* spp., the resistance to echinocandins remains low (< 3% out of the clinically isolated) [14]. An exception is *C. glabrata* and the emerging pathogen *C. auris*, which show a higher percentage of resistance that varies according to the center [8,14]. Remarkably, many clinical isolates of *C. glabrata* and *C. auris* display cross-resistance between echinocandins and azoles. This finding allows explaining their growing presence as fungal bloodstream pathogens.

Of note is the fact that the action mechanism of some antifungals is certainly more complex than initially believed. Thus, AMB generates an intracellular oxidative stress throughout the formation of endogenous reactive oxygen species (ROS) in diverse pathogenic yeasts, which is part of its toxic effect [16]. In *C. albicans*, it has been reported that AMB also triggered the accumulation of the disaccharide trehalose and increased the antioxidant enzymatic activities [17]. In this context, it should be remarked that the Mitogen-Activated Protein (MAP) kinase Hog1 plays a pivotal role in the physiology and virulence of *C. albicans* [18,19], which encompasses the response to oxidative and osmotic stress or the virulence and susceptibility to phagocytic cells [18,19,20,21,22]. Notably, we just demonstrated that the *C. albicans* response to AMB is mediated by the Hog1 pathway [23].

Apart from AMB, other antifungals, structurally unrelated, cause a similar effect [24]. Belenky and co-workers reported that some compounds with fungicidal activity induce a common oxidative-damage cellular death pathway, suggesting that at least part of the fungicidal effect depends on the ROS production [24]. Regarding the echinocandin caspofungin (CAS), it was reported that it also induces the expression of genes associated with the oxidative stress response and activates oxidative stress signaling mediated by Hog1 or Cap1 in *C. albicans* [25]. In addition, ROS derived from the endoplasmic reticulum also contributes to the toxicity of the cell wall stress agent Calcofluor White on *C. albicans*, establishing an additional link between ROS production and cell wall integrity [26]. In contrast, ROS formation and the induction of antioxidant enzymatic activities appear to play a minor role upon Micafungin (MF) treatment [27]. Nevertheless, this antifungal causes morphological alterations and relevant cell wall damage according to its canonical mechanism of action (inhibition of β-(1,3)-glucan synthesis) [27]. CAS and MF display a fungicidal effect as a function of the concentration. There are also differences for each echinocandin concerning in vitro antifungal activity, pharmacokinetics, and toxicity. However, both drugs are currently applied in the clinical therapy of candidemia and other manifestations of invasive candidiasis [28].

In this work, we analyze in depth the putative role exerted by the MAP kinase Hog1 in the response of *C. albicans* to the fungicidal action of MF. We provide consistent evidence in support that MF exposure does not result in the generation of an inner ROS-mediated oxidative stress. We included AMB in the analyses, showing that the polyene is able to induce a Hog1-independent oxidative stress response. Our data also point to the involvement of the MAPK Mkc1 as a main target of the toxic effect triggered by MF.

## 2. Materials and Methods 

### 2.1. Yeast Strains and Growth Conditions

This work was performed with the following *C. albicans* strains: RM100 as the wild type (*ura3::imm434/ura3::imm434 his1::hisG/his1::hisG-URA3-hisG*) [29], the *hog1* null mutant (CNC13, *ura3::imm434/ura3::imm434 his1::hisG/his1::hisG hog1::hisG-URA3-hisG/hog1::hisG*) [30], and the Hog1 reintegrant strain named as Hog1+ (CNC15-10, *ura3::imm434/ura3::imm434 his1::hisG/his1::hisG hog1::hisG/hog1::hisG LEU2/leu2::HOG1URA3*) [29]. The construction of the set of *hog1* mutants tested in Appendix A is depicted elsewhere [25]. 

Yeasts were routinely grown at 37 °C with shaking in 1% YPD (yeast extract, Conda/Pronadisa, Madrid, Spain), 2% Peptone (Conda/Pronadisa, Madrid, Spain), and 2% Dextrose ((D(+) glucose anhidra (PanReac AppliChem, Barcelona, Spain) medium, and the cells were harvested in the exponential phase. For this, overnight cultures were refreshed in new prewarmed YPD to an Optical Density (OD) = 0.05–0.1 measured at 600 nm (OD_600nm_) and further incubated until they reached OD_600nm_= 0.8–1.0. These liquid cultures were used to perform stress sensitivity assays on solid media. Ten-fold cell dilutions from 2 × 10^7^ to 2 × 10^4^ cells/mL were prepared, and 5 µL were spotted on YPD plates supplemented with the indicated concentration of specific compounds. Plates were incubated at 37 °C for 24 h. 

### 2.2. Minimum Inhibitory Concentration (MIC)

The lowest concentration of MF that required inhibiting ≥ 50% of the fungal growth was determined following the procedures indicated by the Clinical & Laboratory Standards Institute (CLSI) [31]. Briefly, a total of 10^3^ yeasts were grown in a 96-well plate containing Roswell Park Memorial Institute medium (RPMI) 1640 (without glutamine, 0.2% glucose, and buffered at pH 7.0 with 0.164-M morpholinepropanesulfonic acid (MOPS) (Lonza Bioscience, Verviers, Belgium) with 2-fold dilutions of MF ranging from 0.004 to 4 mg/L. The reading was performed by spectrophotometry at 490 nm after incubation for 24 h at 37 °C. 

### 2.3. Preparation of Cell-Free Extracts

After exposure to different stresses, samples from the cultures were harvested and resuspended at known densities (10–15 mg/mL wet weight) in the extraction buffer, 100-mM 4-morpholine-ethanesulfonic acid (MES) pH 6.0, containing 5-mM cysteine and 0.1-mM phenyl methyl sulphonyl fluoride (PMSF). The cellular suspensions were transferred into small precooled tubes (1.0-cm diameter) with 1.5-g Ballotini glass beads (0.45-mm diameter). Cells were broken by vigorously vibrating the tubes in a vortex mixer. The tubes were quickly cooled in ice. The crude extract was then centrifuged at 10,000× *g* for 5 min, and the pellet was resuspended in the same buffer at the initial density. For antioxidant assays, the supernatant fraction obtained was filtered through Sephadex G-25 NAP columns (Amersham Pharmacia Biotech AB) previously equilibrated with 50-mM K-phosphate buffer, pH 7.8, to remove low molecular weight compounds.

### 2.4. Enzymatic Assays

Catalase activity was determined at 240 nm by monitoring the removal of H_2_O_2_, as described elsewhere [26]. Measurements of superoxide dismutase (SOD) were carried out spectrophotochemically by the ferricytochrome C method using xanthine/xanthine oxidase as the source of O_2_.^-^ radicals [26]. Data of enzymatic activity were normalized in relation to a control measurement (100%).

### 2.5. ROS and Mitochondrial Membrane Potential Determination

Intracellular ROS were measured with dihydrofluorescein diacetate (DHF), following the procedure described previously [32]. Briefly, *C. albicans* overnight cultures were collected, washed, and resuspended in phosphate-buffered saline (PBS) (BD FACS Flow, from Becton-Dickinson Biosciences, Madrid, Spain) at 10^7^ cells/mL. The strains were treated with MF or AMB to the concentrations indicated for 1 h. Then, DHF was added to the samples to a final concentration of 40 µM and incubated at 37 °C for 30 min. Mitochondrial membrane potential was also determined by flow cytometry using Rhodamine 123 to a 20-µM final concentration and incubated for 30 min at 37 °C. After treatment with Rhodamine 123, the cells were washed twice with PBS to remove the excess of fluorochrome. An EPICS XLMCL4 cytometer (Beckman Coulter, High Wycombe, UK) was used to determine the fluorescence intensity. The analysis of green (FL1) fluorescence intensity was made on a logarithmic scale after gating the intact cells on a forward scatter (FSC) vs. side scatter (SSC) dot plot. The fluorescence histograms correspond to 5000 cells. Data acquisition and analysis were performed using Flowing Software (http://flowingsoftware.btk.fi/).

### 2.6. Real-Time Quantitative RT-PCR (qRT PCR) Analysis

Exponential *C. albicans* cultures growing in YPD at 37 °C were split on three subcultures; then, antifungal treatment was added and samples taken at different time points. The “mechanical disruption” protocol and RNeasy mini kits with column DNase treatment (Qiagen) were used to isolate the total RNA. Then, the amount of RNA was quantified spectrophotometrically, and 2 μg of total RNA were reverse-transcribed by the Superscript First-Strand synthesis system for RT-PCR (Invitrogen). Then, qPCR was performed using the SYBR-green dye method with *ACT1* transcript as the internal standard. Samples and data analyses were run using the Applied Biosystems 7500 Real-Time PCR System. The PCR system was programmed to 95 °C for 10 min, then 95 °C for 14 s and 60 °C for 1 min; the last two steps were repeated 40 times. All samples were analyzed in triplicate, normalized to the *ACT1* gene expression level, and the results were expressed as a fold induction compared to untreated controls for each strain. The primers used in the assay are listed in Table 1.

## 3. Results

### 3.1. Homozygous Disruption of HOG1 Gene does not Increase the Susceptibility to MF

In order to elucidate if Hog1-mediated oxidative stress plays any significant role in the MF action mechanism against *C. albicans*, we analyzed the effect of this echinocandin on a *hog1* null mutant, previously reported to be hypersensitive to oxidants [29]. Firstly, the MIC values were determined for the parental (RM100), the isogenic homozygous mutant *hog1*, and the reintegrant Hog1^+^ strains following the CLSI protocol, being the MIC_50_ for MF identical in all cases (0.016 mg/L). Then, the percentage of cell viability after treatment with 0.05 mg/L MF during 1h at 37 °C was quantified by colony-forming units (CFUs) counting. No significant differences were detected among the wild-type, *hog1*, and Hog1^+^ reintegrant strains (Figure 1a). There were also no differences among these three cell types in spots assays performed on MF-supplemented plates (Figure 1b). Control samples running in parallel confirmed the sensitivity of *hog1* cells to AMB (Figure 1b). Furthermore, we also tested the MF-induced fungicidal effect in a collection of *hog1* mutants constructed in distinct *C. albicans* genetic backgrounds with consistent results (Appendix A). These data strongly support that the implication of the HOG pathway in the *C. albicans* response to MF is essentially different to that recently reported against AMB, which was introduced here as the control.

### 3.2. Differential Action of MF vs. AMB on Intracellular ROS Production and Mitochondrial Membrane Potential

The cell viability determinations upon MF exposure were complemented with flow cytometry assays performed to quantify both the intracellular ROS production and mitochondrial membrane potential. Wild-type and *hog1* mutant cells were treated with MF and AMB for 1 h at 37 °C in PBS buffer, and ROS production was measured by flow cytometry using DHF (Figure 2). Differences were recorded in the function of the strain and the treatment applied. Thus, the intracellular content of ROS triggered by exposure to AMB was higher in the parental strain than in the isogenic *hog1* mutant for the two concentrations of AMB tested (Figure 2). Remarkably, the intracellular ROS formation was similar at both concentrations of AMB, although the percentage of cell survival (propidium iodide, PI plus cells) was not significantly affected at the lowest concentration (Appendix A). In turn, only the largest doses of AMB provoked a noticeable degree of cell killing upon PI staining, which was particularly evident in *hog1* null cells (Appendix A). In contrast, no significant difference in both the endogenous ROS content (Figure 2) and the percentage of cell killing (Appendix A) could be observed after MF treatment (0.05 mg/L) with respect to the measured basal levels in both strains. 

The mitochondrial membrane potential was also quantified for RM100 and *hog1* null strains upon antifungal exposure (Figure 3). Once again, despite the higher basal level recorded in *hog1* cells, the mitochondrial membrane potential showed a larger increase in parental cells with respect to the *hog1* mutant exclusively after AMB treatment but not in the presence of MF (Figure 3). These results indicate that AMB promotes a marked polarization of the mitochondrial membrane, while MF lacks this effect. Moreover, although the absence of a functional Hog1 seems to induce in *C. albicans* the capacity to produce high levels of both endogenous ROS and mitochondrial membrane potential, these Hog1-defective cells always reached lower values of these two parameters than their parental cells (Figure 2 and Figure 3). 

### 3.3. Role of AMB and MF on Antioxidant Gene Expression and Enzymatic Activities

Since AMB activates several antioxidant enzymatic activities in *C. albicans* [17], we aimed to analyze the putative role played by Hog1 in the gene expression and enzymatic activity of two main antioxidant enzymes, i.e., catalase and superoxide dismutase (SOD) in the *C. albicans* response to MF and AMB. We initially determined the level of mRNA expression for *CAT1* and *SOD2* genes upon exposure to toxic concentrations of both compounds by RT-qPCR at different times after the antifungal addition. AMB induced a significant increase of *CAT1* mRNA transcript after 30 min in the three strains analyzed (Figure 4). In addition, AMB increased the expression of *SOD2* mRNA that was detected at 15 min and maintained until 60 min, reaching a peak at 30 min (Figure 4). Notably, the gene expression pattern was similar for the three analyzed strains, although the *hog1* mutant displayed a higher degree of expression (30 min) than the parental and reintegrant strains. In turn, MF failed to induce a conspicuous antioxidant gene expression under all treatments accomplished (Figure 4). These data strongly point to a differential and specific transcriptional response triggered by AMB and MF. Furthermore, Hog1 may control the expression level of certain antioxidant genes. 

Catalase and superoxide dismutase activities were also quantified in wild-type, *hog1*, and reintegrant strains exposed to MF (0.05 mg/L) or AMB (0.25 mg/L) for 1 h (Figure 5). AMB induced a clear activation of catalase and SOD activities, which was not observed in the same cells exposed to MF (Figure 5). Once more, these results evidenced that AMB and MF exert distinct effects on the antioxidant protective system of *C. albicans* cells. The level of catalase and SOD dismutase activities induced by AMB were consistently higher in the *hog1* mutant than in the parental and reintegrant strains (Figure 5). These higher antioxidant activities closely correlate with the above-reported ROS production and enhanced antioxidant gene expression (Figure 2, Figure 3 and Figure 4). Then, *hog1* cells seem able to respond to AMB by mounting a detoxifying response that may counteract the generation of intracellular ROS. In spite of the induction of a strong antioxidant response, this does not ensure an improvement of cell survival.

### 3.4. The Presence of AMB and MF is Sensed through Different Signaling Pathways in C. albicans

Our data clearly indicate that AMB and MF exert dissimilar effects in *C. albicans* cells and that the MAPK Hog1 is mostly relevant for cell survival in the presence of AMB. In order to get a better understanding of the action of these antifungals, the activation degree of three MAPK proteins involved in the response to different stresses was studied. Exponentially growing cultures were exposed to 0.05 mg/L MF or 0.25 mg/L AMB, and identical samples were taken at different time points for immunoblotting analyses. According to the results presented in Figure 6, the addition of AMB triggered Hog1 phosphorylation after five min, which was further maintained during at least 60 min (Figure 6). The MAPK Mkc1, an element of the cell wall integrity pathway [33], was also phosphorylated in the presence of AMB. This phosphorylation was shorter in time, since it was only detected at 5 to 15 min post-AMB exposure. Notably, MF also triggered a more intense Mkc1 phosphorylation, which was observed in the interval from 5 min to 60 min after the antifungal supply (Figure 6). In conclusion, the presence of external AMB and MF seem to be detected by *C. albicans* through different MAPK signaling pathways. The response triggered by AMB resembles the phosphorylation pattern triggered by other oxidative agents, such as H_2_O_2_ [34,35] or arsenite [36]. Meanwhile, MF, as other echinocandins, provoked the alteration of the cell wall. This action was then followed by the activation of the cell wall integrity pathway mediated by Mkc1. Remarkably; no significant Cek1 phosphorylation was detected in response to MF, despite the fact that this MAPK mediates cell wall biogenesis [37].

## 4. Discussion

Morbidity and mortality caused by invasive fungal infections are increasing in hospitalized patients and immunocompromised individuals in developed countries [7,8,38]. *C. albicans* remains as the most common fungal pathogen causing nosocomial infections, although other *Candida* spp. are becoming important [4,5]. The antifungal arsenal has long-been quite limited, mainly formed by polyenes (AMB belongs to this antifungal family) and azoles. This worrying fact joined to the elevated failure rates, the significant adverse effects, the rise of resistant isolates, and the emergency of new pathogenic fungal species [39,40] led to the development of a new family of antifungals, named echinocandins. These antifungals inhibit the β-1,3-D-glucan synthase, interfering with the synthesis of the fungal cell wall [41]. Different studies have shown that the fungicidal efficacy of echinocandins is comparable with that of AMB [42,43], and therefore, their use in hospital has become widespread. The introduction of echinocandins as first-line treatments against nosocomial candidemias has decreased the mortality attributable to these fungal infections, although it remains substantial [13]. Three echinocandins are currently used in clinical practice: Caspofungin (CAS), Micafungin (MF), and Anidulafungin, which possess fungicidal in vitro and in vivo activity against *Candida* and *Aspergillus* spp. These compounds are well-tolerated in medical trials and are administered parenterally [44].

Apart from the canonical mechanism of action of the echinocandins, it has been reported that CAS triggers the translocation of Cap1 and Hog1 to the nucleus, indicating that both pathways (involved in the oxidative stress response) become activated in response to this antifungal [25]. Indeed, *C. albicans* cells respond to CAS, increasing both the gene expression of antioxidant enzymes, as well as their corresponding catalytic activity [25]. These observations suggest that oxidative stress can contribute to the fungicidal action of CAS, as it has been suggested for other antifungals, including AMB [16,24,27,45]. Certainly, CAS induces apoptosis at concentrations lower than or equal to the MIC in *C. albicans,* which is preceded by ROS induction and mitochondrial membrane potential dissipation [46]. Here, we demonstrate that MF acts in a different way; our experimental evidence with MF strongly supports that this antifungal does not induce ROS formation or increase antioxidant enzyme activities or gene expressions (Figure 2, Figure 4 and Figure 5) [27], albeit MF exhibits fungicidal activity against *C. albicans*. Supporting these data, CAS was the only echinocandin able to induce mitochondrion-derived ROS in the filamentous fungus *Aspergillus fumigatus* [47] and to undergo conformational changes in low ionic solution with a subsequent increase of its toxicity, attributed to a higher ROS production, in multidrug resistance *Candida* and bacteria cells [48]. In order to fortify our conclusions, we made use of mutants defective in the MAPK Hog1, which are hypersensitive to oxidants [23]. The percentage of cell survival, as well as the genetic and biochemical indicators of oxidative stress analyzed in the three isogenic strains (Hog1 parental, *hog1* homozygous disrupted, and Hog1^+^ reintegrant), failed to be responsive to the exposure with toxic doses of MF (Figure 1, Figure 2, Figure 3, Figure 4, Figure 5 and Appendix A). Therefore, our results strongly support the hypothesis that MF sensitivity is independent of the MAPK Hog1 and does not generate oxidative stress in *C. albicans*. 

Side effects of the MF treatment include hemolytic anemia and thrombocytopenia in humans. Peter and co-workers demonstrated that human erythrocytes exposed to MF did not induce ROS production or activation of the oxidative stress signaling pathway [49]. In conclusion, MF neither causes oxidative stress in human erythrocytes nor in *C. albicans* cells and, probably, in other cell models. 

Additionally, MF triggers Mkc1 phosphorylation, while other important MAPK pathways, such as Cek1 and Hog1, do not become phosphorylated (Figure 6). Mkc1 phosphorylation can be induced in response to cell wall-disturbing compounds or under oxidative stress challenge [34]. Since this antifungal does not increase intracellular ROS accumulation (Figure 2) nor induces antioxidant gene expression or antioxidant enzyme activity (Figure 4 and Figure 5), Mkc1 phosphorylation must be likely due to cell wall perturbations. Indeed, MF caused severe cell wall damage [27], which could trigger Mkc1 activation (Figure 6). The study of defective mutants constructed in the *MKC1* gene and other elements involved in this signaling pathway can elucidate the relevance of the Mkc1 pathway in sensing MF. In this way, our results point out that Hog1 should not play an important role in MF signaling or sensitivity, although this MAP kinase pathway has also been implicated in cell wall biogenesis [50].

On the other hand, AMB is able to induce a clear oxidative stress and Hog1 signaling plays a relevant role in the response to this antifungal in *C. albicans* (Figure 2 and Figure 3) [23]. Remarkably, the mean fluorescent intensity of intracellular ROS in AMB-treated cells was lower in *hog1* than in parental cells in spite of the lower MICs displayed by the mutant (Figure 2). In turn, the lower levels of both intracellular ROS and mitochondrial membrane potential displayed by *hog1* cells correlate with a higher mRNA expression of *CAT1* and *SOD2* accompanied by the increase of the corresponding catalase and SOD activities compared to RM100 cells (Figure 4 and Figure 5). These detoxifying enzymes may counterbalance the oxidative stress generated by AMB, but they fail in preventing the loss of cell viability. 

The response to oxidative stress in *C. albicans* appears to be quite complex. It involves the Hog1 pathway but, also, the Mkc1-mediated pathway and different transcription factors, like Cap1, Skn7, or Pho4 [51,52]. Transcriptomic analyses revealed that *hog1* mutants were able to respond to oxidant treatments regardless of its evident sensitivity [21]. These studies were exclusively performed with H_2_O_2_, but they could be extended to other oxidative agents, including the antifungals used in daily medical practice. 

Here, we demonstrated that neither the MAPK Hog1 nor the generation of an internal oxidative stress were involved in the signaling to MF and the further response in *C. albicans*. These results evidence that not all the compounds with fungicide activity cause oxidative stress, MF being an example. This fact can be interesting, since part of the side effects caused by antifungal compounds is due to their ability to trigger oxidative stress.

## Figures and Tables

**Figure 1 microorganisms-08-01867-f001:**
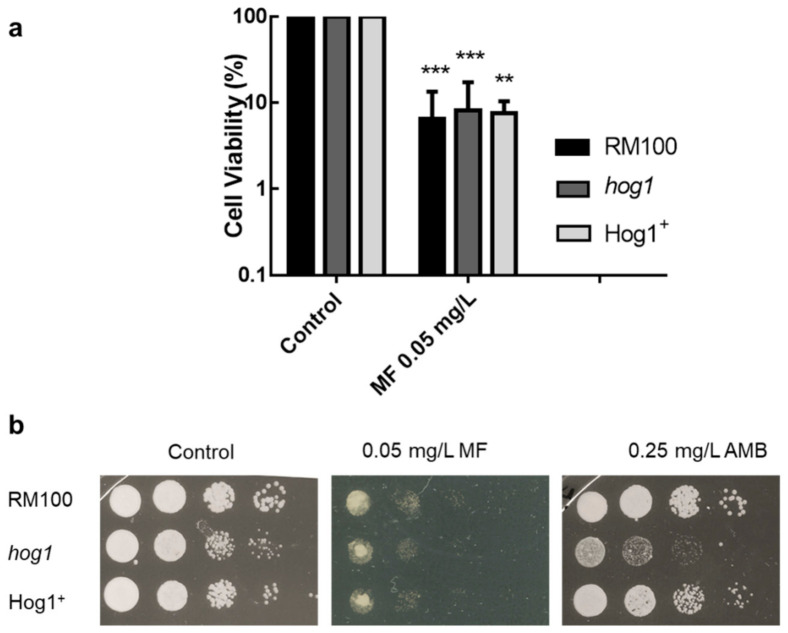
Sensitivity displayed by the *hog1* null mutant to antifungal exposure with micafungin (MF) and osmotic stress. (**a**) Strains RM100 (black histograms), *hog1* (dark-grey histograms), and Hog1^+^ (light-grey histograms) were treated with 0.05-mg/L MF or 0.25 mg/L Amphotericin B (AMB) for 1 h in liquid yeast extract peptone dextrose (YPD) at 37 °C. The cells were then collected and spread on YPD plates, and cell viability was determined by colony-forming unit (CFU) counting. The data shown are the mean ± SD of three independent experiments. ** *p* < 0.01 and *** *p* < 0.001 represent statistically significant differences with respect to an untreated control, according to the Mann Whitney U test. (**b**) Ten-fold cell suspensions were spotted on YPD plates containing MF and AMB. The plates were incubated at 37 °C for 24 h.

**Figure 2 microorganisms-08-01867-f002:**
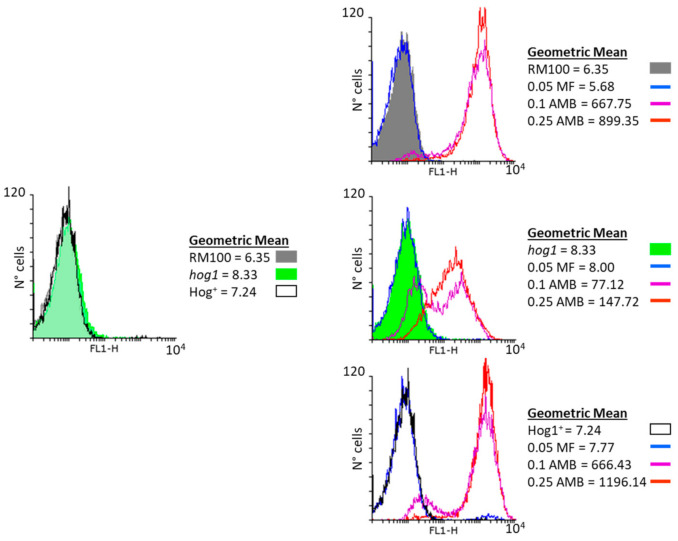
Quantification of intracellular ROS formation after treatment with MF and AMB. Exponential YPD-grown cells were incubated in phosphate-buffered saline (PBS) in the presence of 0.05 mg/L MF, 0.1 mg/L AMB, or 0.25 mg/L AMB at 37 °C. After 1 h, the level of intracellular ROS was quantified by flow cytometry using dihydrofluorescein diacetate (DHF) (40 µM final concentration). Histogram on the left shows basal intracellular ROS of RM100 (filled dark-grey histograms), *hog1* mutant (filled green histograms), and Hog1^+^ reintegrant (filled white-black line histograms). Histograms represent the total cell number with respect to the green (FL1) fluorescence intensity. The data shown are representative of three independent experiments. The geometric means of the fluorescence intensity are indicated in the text box for ROS measurements.

**Figure 3 microorganisms-08-01867-f003:**
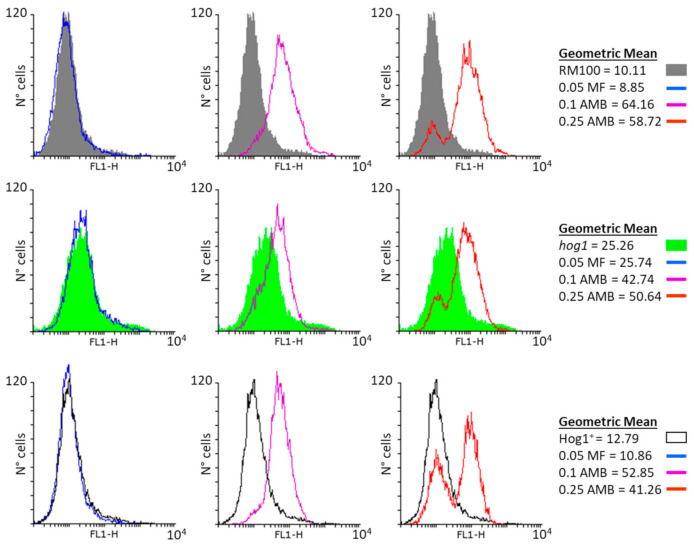
Effect of the antifungal treatment on the mitochondrial membrane potential in RM100, *hog1* mutant, and Hog1^+^ reintegrant strains. Yeast cells were treated with MF (0.05 mg/L) and AMB (0.1 and 0.25 mg/L) in PBS for 1 h at 37 °C. Then, they were incubated with 20 µM Rhodamine 123 for 30 min, and the membrane potential was determined by flow cytometry. The upper row shows the mitochondrial membrane potential of RM100 untreated (basal level) and after antifungal treatments. The row in the middle shows the mitochondrial membrane potential of the *hog1* mutant untreated (basal level) and after antifungal treatments and the lower row shows the mitochondrial membrane potential of the Hog1^+^ reintegrant strain. Histograms represent the cell number with respect to the green (FL1) fluorescence intensity. The data shown are representative of three independent experiments. The geometric means of the fluorescence intensity are indicated in the text box. Filled gray histograms belong to the basal wild-type RM100 strain, filled green histogram belongs to the *hog1* mutant, and white-filled black lines belong to the basal Hog1^+^ reintegrant. For other details, see Figure 2.

**Figure 4 microorganisms-08-01867-f004:**
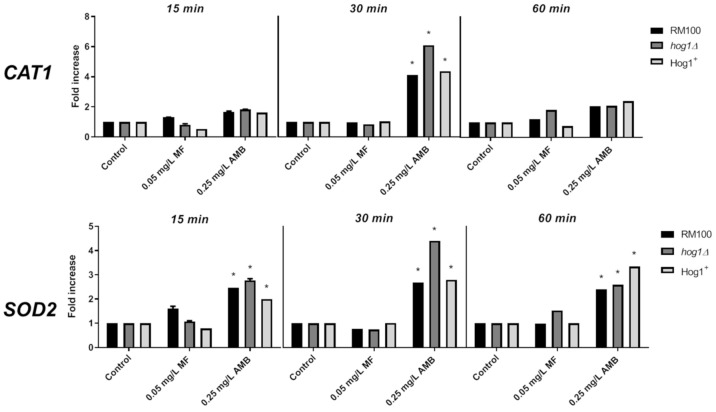
Level of *CAT1* and *SOD2* mRNA transcripts in response to AMB and MF. Exponentially growing cells were treated with 0.25 mg/L AMB or 0.05 mg/L MF at 37 °C. Samples were collected at 15, 30, and 60 min and processed for RT-qPCR analysis. *ACT1* mRNA transcript was used as the internal control, and *CAT1* and *SOD2* mRNA transcripts were detected using specific primers. The data shown are the mean ± SD of three independent experiments and normalized to untreated cultures at the same time point for each strain. * Two-fold induction or more was considered significant.

**Figure 5 microorganisms-08-01867-f005:**
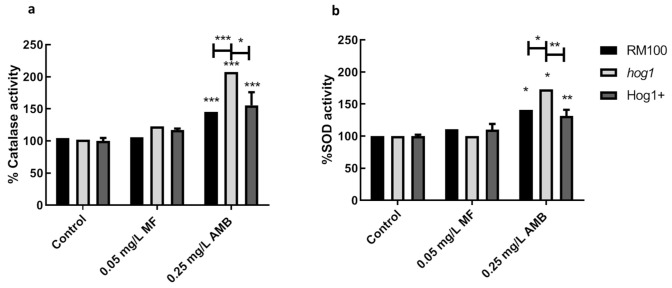
(**a**) Catalase and (**b**) superoxide dismutase (SOD) activities after treatment with MF and AMB. Cells were incubated in YPD with MF and AMB to the indicated concentration at 37 °C for 1 h. Cytosolic extracts were obtained, and the activities were determined as it is indicated in Materials and Methods. The data show the mean ± SD of three independent experiments. **p* < 0.05, ***p* < 0.01, and ****p* < 0.001 represent statistically significant differences with respect to an untreated control according to the Mann Whitney U test.

**Figure 6 microorganisms-08-01867-f006:**
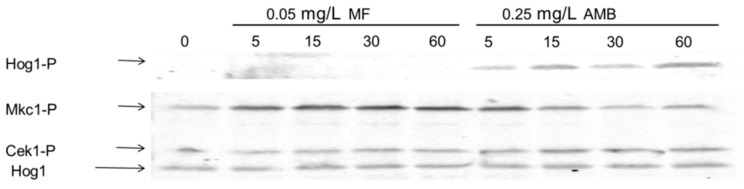
Role of the Mitogen-Activated Protein kinase (MAPK) pathways in the response to MF and AMB. Wild-type cells were challenged with MF or AMB at 37 °C in YPD liquid medium. Then, samples were collected at different time points (indicated in min) and processed for immunoblotting. The Hog1 phosphorylated form was detected using anti-phospho-p38 antibody (Hog1-P), and phosphorylated Mkc1 (Mkc1-P) and Cek1 (Cek1-P) forms were detected using anti-phospho-p42-44 antibody. Total Hog1 protein was detected with an anti-ScHog1 antibody and used as the load control.

**Table 1 microorganisms-08-01867-t001:** Primers used for RT-qPCR analysis.

Gene	Primer’s Name	Sequence
***ACT1***	o-ACTQTup	TGGTGGTTCTATCTTGGCTTCA
	o-ACTQTlw	ATCCACATTTGTTGGAAAGTAGA
***CAT1***	o-CAT1up-QT	ATCCCAGTGAACTGTCCTGTCA
	o-CAT1lw-QT	ACCATTAACAGTCATAGCACCATCTCT
***SOD2***	o-SOD2up-QT	TGCTTCCAAGACTTTCACTAGATCTAA
	o-SOD2lw-QT	TGGTTCAGTAGCGGAGAATTCAT

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
