# Peer review of "The Fungicidal Action of Micafungin is Independent on Both Oxidative Stress Generation and HOG Pathway Signaling in Candida albicans"

_microorganisms, 2020, doi:10.3390/microorganisms8121867_

Round 1

Reviewer 1 Report

Overall impression

The authors describe a series of experiments that show that the action of a glucan synthase inhibitor, micafungin, does not cause oxidative stress in C. albicans. The team goes on to show that the micafungin response does not depend on the action of the MAPK Hog1. The experiments are well-described and the execution is solid. It is, however, not clear how the authors derived their original hypothesis of oxidative stress being a part of micafungin action. Furthermore, since the study is in contradiction with earlier findings on the action of echinocandins, it should have included a caspofungin control to allow for a better comparison of competing findings.

Major issues

The premise of the study to examine the effects of Micafungin on the generation of radical stress in C. albicans, and to describe the role of HOG signaling in the radical stress mitigation. The reported outcomes lead the authors to conclude, accurately, that there is no radical stress in Micafungin-treated C. albicans and that HOG signaling does not play a role in resistance. The reader is left questioning the rationale of the study – why should there be radical stress when cell wall glucan synthesis is inhibited?

The justification for the study is a previous report that a related cell wall synthesis inhibitor, caspofungin, does generate radical stress. The results of the present study seem to contradict these findings.  It would have been important to include a caspofungin control in the present study to ensure that the surprising caspofungin findings can be reproduced. Instead, the authors chose to control with an entirely unrelated antifungal drug, amphothericin B, whose mechanism of action as a membrane disruptor provides a more intuitive explanation of radical stress.

The conclusion that not all antifungal drugs act via the generation of radical stress is thus somewhat puzzling, given that there was no reason to suspect such a correlation in the first place.

Minor issues

The manuscript has been thoroughly proofed and meets the standards for clear English. Figure 3 still has an axis label in Spanish.

Author Response

Introduction:

  1. The introduction should greatly benefit from describing the differences between caspofungin and micafungin. Is it known why caspofungin triggers oxidative stress and micafungin not? How structurally are they different? Please explain. 

We have included some information along the introduction and in the discussion section to address these questions:

Line 80:

In addition, ROS derived from the endoplasmic reticulum also contributes to the toxicity of the cell wall stress agent Calcofluor White on C. albicans establishing an additional link between ROS production and cell wall integrity

Line 85 “CAS and MF display a fungicidal effect as a function of the concentration. There are also differences for each echinocandin concerning in vitro antifungal activity, pharmacokinetics and toxicity. However, both drugs are currently applied in clinical therapy of candidemia and other manifestations of invasive candidiasis [27]

Line 325 (discussion):” Three echinocandins are currently used in clinical practic: Caspofungin (CAS), Micafungin (MF) and Anidulafungin with possess fungicidal in vitro and in vivo activity against Candida and Aspergillus spp. These compounds are well tolerated in medical trials and are administered parenterally [42]”

Line 340 (discussion): “Supporting these data, CAS was the only echinocandin able to induce mitochondrion-derived ROS in the filamentous fungus Aspergillus fumigatus [47] and to undergo conformational changes in low ionic solution with a subsequent increase of its toxicity, attributed to a higher ROS production, in multidrug resistance Candida and bacteria cells (Sumiyosho et al 2020).”

  1. Line 46: azoles do not target ergosterol but it targets the biosynthesis of ergosterol by inhibiting a key enzyme in its biosynthesis. Please clarify.

The reviewer is right, although both antifungals act on the plasma membrane the mechanism of action is different. We have rephrased this part as follows:

“The cellular target of AMB is the ergosterol present in the fungal plasma membrane and mitochondrial membranes. AMB binds ergosterol forming pores and altering selective permeability [9,10]. On the other hand, azoles halt ergosterol biosynthesis by the specific inhibiting of 14α-demethylase (Erg11) leading to depletion of ergosterol and accumulation of toxic sterol precursors [10].

  1. Line 49: please add how glucan synthesis is inhibited by echinocandins? Echinocandins inhibit FKS complex of beta (1-3- glucan synthase), an important component in its biosynthesis

Following the reviewer suggestion, we have included the following sentence : Resistance to echinocandins are mainly due to mutations on “hot spot” regions of FKS1and FKS2 which encode the β-(1, 3)-glucan synthase [14] meanwhile, resistance to azoles involve different and non-exclusive mechanisms  [15]”. .Moreover, we have Included the references suggested by the reviewer. 

https://doi.org/10.1093/cid/civ791
https://doi.org/10.3390/antibiotics9060312

  1. Line 52: how common is echinocandin resistance in clinical isolates of candida?

The following information has been included in the introduction:

“In Candida albicans and other Candida spp the resistance to equinocandins remains low (<3% out of clinical isolated) [14]. An exception is C. glabrata and the emerging pathogen C. auris, which show a higher percentage of resistance that vary according to the center [8,14]. Remarkably, many clinical isolates of C. glabrata and C. auris displayed echinocandins and azoles cross –resistance. This finding allows to explain their growing presence as fungal bloodstream pathogen.”

  1. Line 55: "Thus, recent finding...." should be "Thus, a recent finding..."

The error has been corrected

Materials and Methods:

  1. The authors have stated that they did MICs. MIC should be defined as the "minimum inhibitory concentration" in the materials and methods. How were the MICs performed. Please describe it in the Materials and Methods. Also MIC50 was analyzed whereas the standard for MIC analysis is usually MIC80. Please explain.

A description of the MIC determination has been include in material and methods.. The lowest concentration to inhibit ≥ 50% of the fungal growth was used following the CLSI guidelines for echinocandins.

  1. The authors performed spot assays in Figure 1b. Please provide in the materials and methods how the spot assays were done. How many cells from each strains were use? what dilutions were used?

The following sentences have been included in material and methods.

“These liquid cultures were used to perform stress sensitivity assay on solid media. Ten-fold cell dilutions from 2.107 to 2.104 cel/mL were prepared and 5 µL were spotted on YPD plates supplemented with the indicated concentration of specific compounds. Plates were incubated at 37ºC for 24h.”

  1. The materials and methods should also reflect the RT-qPCR conditions used in the experiments. Also, please keep the names of RT-qPCR consistent throughout the text. Sometime it is designated as "Q-PCR.

The conditions used to amplified the DNA transcript have been included in the text: “The PCR system was programmed to 95ºC for 10 minutes, then 95ºC for 14 seconds and 60ºC for 1 minute, last two steps are repeated 40 times”

Results and Discussions:

  1. Throughout the text in many places C. albicans is not italicized. For example line 132. 
  2. Gene names also needs to be italicized for example line 205.

I am sorry for these mistakes, I must have made a mistake when adapting the final format, all the italics were gone in this section. Now, I have reviewed the whole test.

  1. Figure 1a: Y axis title : Please explain what is "Viabilidad"? Did the the authors mean viable?

The X axis title has been changed to cell viability

  1. Figures 2 and 3: Since the re-integrant of  hog1 was already available in this study, why were it not used in these analyses? A revertant should always be use for all analyses. Please add the data for the revertant.

Following the reviewer suggestion  data form Hog1+ reintegrant strain have been included in figures 2 and 3

  1. Figure 2: Is the mean for MF 10.41 or 10,41? There should be a dot instead of comma between the numbers.

The legend of the figure 2 has been corrected

Reviewer 2 Report

The article by Monge et al. describes how the fungicidal activity of micafungin is independent of oxidative stress and the HOG pathway signaling. Though the paper interestingly highlights how micafungin activity is different from caspofungin and amphotericin B, here are my major concerns about this article;

Introduction:

  1. The introduction should greatly benefit from describing the differences between caspofungin and micafungin. Is it known why caspofungin triggers oxidative stress and micafungin not? How structurally are they different? Please explain.  
  2. Line 46: azoles do not target ergosterol but it targets the biosynthesis of ergosterol by inhibiting a key enzyme in its biosynthesis. Please clarify.
  3. Line 49: please add how glucan synthesis is inhibited by echinocandins? Echinocandins inhibit FKS complex of beta (1-3- glucan synthase), an important component in its biosynthesis
    https://doi.org/10.1093/cid/civ791
    https://doi.org/10.3390/antibiotics9060312
  4. Line 52: how common is echinocandin resistance in clinical isolates of candida?
  5. Line 55: "Thus, recent finding...." should be "Thus, a recent finding..."

Materials and Methods:

  1. The authors have stated that they did MICs. MIC should be defined as the "minimum inhibitory concentration" in the materials and methods. How were the MICs performed. Please describe it in the Materials and Methods. Also MIC50 was analyzed whereas the standard for MIC analysis is usually MIC80. Please explain.
  2. The authors performed spot assays in Figure 1b. Please provide in the materials and methods how the spot assays were done. How many cells from each strains were use? what dilutions were used?
  3. The materials and methods should also reflect the RT-qPCR conditions used in the experiments. Also, please keep the names of RT-qPCR consistent throughout the text. Sometime it is designated as "Q-PCR.

Results and Discussions:

  1. Throughout the text in many places C. albicans is not italicized. For example line 132. 
  2. Gene names also needs to be italicized for example line 205.
  3. Figure 1a: Y axis title : Please explain what is "Viabilidad"? Did the the authors mean viable?
  4. Figures 2 and 3: Since the re-integrant of  hog1 was already available in this study, why were it not used in these analyses? A revertant should always be use for all analyses. Please add the data for the revertant.
  5. Figure 2: Is the mean for MF 10.41 or 10,41? There should be a dot instead of comma between the numbers.

Author Response

QUESTION:The premise of the study to examine the effects of Micafungin on the generation of radical stress in C. albicans, and to describe the role of HOG signaling in the radical stress mitigation. The reported outcomes lead the authors to conclude, accurately, that there is no radical stress in Micafungin-treated C. albicans and that HOG signaling does not play a role in resistance. The reader is left questioning the rationale of the study – why should there be radical stress when cell wall glucan synthesis is inhibited?

ANSWER :It has been previously reported that different compounds with antifungal activity induce ROS that contribute to the cell death. These compounds are structurally and mechanistically diverse, i.e. AMB, miconazole, ciclopirox and caspofungin. These drugs have in common the fungicidal activity and Belenky and co-workers hypothesize that the fungicidal activity was due to the oxidative stress induction. Since MF is an echinocandin with fungicidal activity against C. albicans. We start from the Belenky´s hypothesis to address the study of the oxidative stress generated by MF. However, previous studies indicated that MF did not induce a clear oxidative because of that we decided to explore the role of Hog1.

QUESTION:The justification for the study is a previous report that a related cell wall synthesis inhibitor, caspofungin, does generate radical stress. The results of the present study seem to contradict these findings.  It would have been important to include a caspofungin control in the present study to ensure that the surprising caspofungin findings can be reproduced. Instead, the authors chose to control with an entirely unrelated antifungal drug, amphothericin B, whose mechanism of action as a membrane disruptor provides a more intuitive explanation of radical stress.

ANSWER: Although the role of Hog1 in the response to caspofungin has been previously reported (doi: 10.3109/13693780802552606), we and others (doi: 10.1128/EC.00081-09 and  doi: 10.1091/mbc.e08-02-0191) have not detected an altered sensibility to caspofungin in the hog1 mutant on spot sensitivity assay. This previous observation led us to look for another positive control against which the hog1 mutant was sensitive. This control was AMB since we have reported recently that the response to AMB was mediated by Hog1 and interestingly, the hog1 mutant is still able to respond to AMB triggering for example trehalose accumulation. In this work, we approach the study of a relatively unknown echinocandin, MF and we further explored the response triggered by AMB in wild type and hog1 mutant strains.

The study of the relationship between the HOG pathway and the Caspofungin has been addressed indirectly by Rauceo and co-workers studying the role of the Sko1 transcription factor, they concluded that Sko1 mediated the response to caspofungin independently of Hog1.

QUESTION:The conclusion that not all antifungal drugs act via the generation of radical stress is thus somewhat puzzling, given that there was no reason to suspect such a correlation in the first place.

ANSWER.If a fungicidal compound generates oxidative stress and trigger a common oxidative-damage cellular death pathway (doi: 10.1016/j.celrep.2012.12.021) it is not unreasonable to think that MF will behave in a similar way. The fact that this compound does not induce oxidative stress indicates that ROS is not an essential element of its fungicidal activity and sheds light on the mechanism of action of MF that differs from Caspofungin. A recent work reports that CAS (and no other echinocandins) has a potent effect against multidrug-resistant (MDR) Candida when dissolved on low ionic solutions. This activity was due to a conformational change and intracellular accumulation that leads to increase ROS production (https://doi.org/10.1038/s41598-020-74749-8).

Some of these reasoning have been included in the discussion of the revised manuscript.

QUESTION: Minor issues

The manuscript has been thoroughly proofed and meets the standards for clear English. Figure 3 still has an axis label in Spanish

ANSWER: The mistakes have been corrected

Round 2

Reviewer 1 Report

The manuscript has been improved significantly; even though no new experiments were performed, the text now addresses my concerns and gives a better rationale for the study. 

Reviewer 2 Report

The authors have answered to all my queries satisfactorily.

Minor Edits:

Line 62: Echinocandin is spelled as equinocandin. Please change.